# Greater Post-Operative Nutrition Risks Identified in Pediatric and Adolescent Patients after Anterior Cruciate Ligament Reconstruction Regardless of Age and Sex

**DOI:** 10.3390/nu16152379

**Published:** 2024-07-23

**Authors:** James J. McGinley, Jessica Dabis, Taylor Morrison, Caroline Podvin, Henry B. Ellis, Sophia Ulman

**Affiliations:** 1Center for Excellence in Sports Medicine, Scottish Rite for Children Orthopedic and Sports Medicine Center, 5700 Dallas Parkway, Frisco, TX 75034, USA; jamesmcginley@att.net (J.J.M.); jessica.dabis@tsrh.org (J.D.); taylor.morrison@tsrh.org (T.M.); ccpodvin@gmail.com (C.P.); henry.ellis@tsrh.org (H.B.E.); 2Department of Orthopaedic Surgery, University of Texas Southwestern Medical Center, 1801 Inwood Road, Dallas, TX 75390, USA

**Keywords:** nutrition, diet, sport, rehabilitation, risk assessment, youth, child, lower extremity, surgery

## Abstract

Systematic detection of risky nutrition behaviors after sports surgery may better promote healing for return-to-sport. The purpose of this study was to assess nutritional behavior differences between patients following anterior cruciate ligament reconstruction (ACLR) and following other lower-extremity orthopedic surgeries. One pediatric sports medicine center was reviewed for a custom Sports Nutrition Assessment for Consultation, which investigates nutrition-related risk factors for youth athletes at their first post-operative visit. Patients reported “Yes” or “No” to eight questions, after which they were offered a nutrition consultation for any response indicating risk. A total of 243 post-ACLR and 242 non-ACLR patients were reviewed. The post-ACLR patients more often reported a change in appetite (*p* = 0.021), recent weight changes (*p* = 0.011), a desire to better understand nutrition (*p* = 0.004), and recommendations to change their body composition (*p* = 0.032). More post-ACLR patients were identified for a nutrition consultation (*p* = 0.002), though an equal percentage accepted the consultation between groups. Age and sex were not determined to be significant confounders after matched sub-analysis. The post-ACLR patients more often reported nutrition risks, specifically weight-related issues, regardless of age or sex. Sports surgeons should regularly inquire about nutrition-related concerns with patients and refer to sports dietitians for recovery nutrition support as needed, particularly after ACLR.

## 1. Introduction

As many as 400,000 individuals undergo an anterior cruciate ligament reconstruction (ACLR) each year [1], with a multi-billion-dollar total cost to society [2]. After surgery, a lengthy rehabilitation process produces variable return-to-sport rates, as 8–23% of athletes sustain a re-injury [3]. Return-to-sport clearance for young athletes involves both functional clearance and psychological readiness [4,5], given that time off from sport may impact an athlete’s mental status [6]. One essential facet of psychological readiness is the topic of nutrition, which bridges an athlete’s mental state and their physiologic requirement for energy upkeep. Unlike adults, however, youth recovering from surgery must account for the energy needs of normal growth in addition to their physiologic requirements for recovery during a period in which body image and social consciousness are growing [7]. The culmination of these factors leaves youth recovering from ACLR or other surgeries in a vulnerable position regarding their rehabilitation success.

Energy intake needs are increased after injury or surgical intervention [8]. After ACLR, relative disuse of the limb is encouraged as a post-operative precaution, and muscle atrophy is an inevitable component of the post-operative sequelae [9,10]. Recovery from post-surgical muscle atrophy and anabolic resistance is crucial to rebuild lower-limb strength and meet the demands of sport [8,10]. Carbohydrate and protein intake are, therefore, essential to stimulate strength gains [8]. However, without a proper understanding of the adolescent athlete population’s current education level and behavior patterns in nutrition, targeted strategies for post-operative behavior change cannot be implemented. To our knowledge, the topic of nutrition as a contribution to physiologic readiness for ACLR return-to-sport has not been systematically evaluated.

Given the high prevalence of ACLRs in the adolescent population, the importance of an intervention-specific understanding of post-operative nutrition is heightened. The systematic detection of risky nutritional behaviors and decreased post-operative competency of caloric needs may better promote healing and muscular development for ACLR return-to-sport, though it is rarely addressed. Addressing nutrition will allow for healthcare providers to develop more appropriate protocols for best practice in rehabilitation management. The purpose of this study was to assess self-reported nutritional behaviors between a post-ACLR patient group and a group who underwent non-ACLR lower-extremity surgery at a pediatric sports medicine center. It was hypothesized that a significant deficit in healthy nutritional habits would be present in athletes who underwent an ACLR due to the demands of a lengthy recovery.

## 2. Materials and Methods

### 2.1. Patients

After approval from the local Institutional Review Board according to the Declaration of Helsinki (STU 062017-100), patients who underwent lower-extremity surgery at a single tertiary sports medicine center were retrospectively reviewed. As this study was retrospective and involved minimal risk to the study subjects, a waiver of informed consent was obtained, given that the research could not practically be carried out otherwise. Patients were included if they were 8–19 years of age at the time of surgery, given that younger patients are less frequently treated surgically at the study institution. The included patients must have completed the institution-specific Sports Nutrition Assessment for Consultation (SNAC) between 30 November 2021 and 19 January 2023 at their first post-operative clinic visit.

The non-ACLR group included a large range of procedures, but the most frequent diagnoses were meniscal tears (*N* = 153), patellar instability/subluxation (*N* = 123), chondral injuries (*N* = 50), and discoid meniscus (*N* = 37). Patients were eligible regardless of the number of concomitant procedures they underwent, but any ACLR assigned the patient to the ACLR group.

### 2.2. Procedures

The institution-specific SNAC survey used in the current study was developed by an experienced sports medicine dietitian in collaboration with other sports medicine staff (Table 1). The survey was designed to identify potential nutrition-related risk factors in youth athletes that might delay the athlete’s recovery or put them at increased risk for re-injury and includes eight questions with “yes” or “no” responses, followed by two female-specific questions that may indicate overtraining or undernutrition. A “yes” response to any question was considered a “risky” nutritional habit and prompted an option to meet with a dietitian for a nutrition consultation during the patient’s visit. The patients’ SNAC responses were collected through standard-of-care electronic intake questionnaires. Age and sex were collected through review of the patients’ electronic medical records.

### 2.3. Statistical Analysis

The cohort was first divided into ACLR and non-ACLR patient groups. Demographic variables were analyzed by Mann–Whitney U-tests after statistically significant results from Shapiro–Wilk normality testing. The impact of ACLR on the frequency of “yes” responses to each SNAC question was analyzed by chi-square tests. To determine the role of sex as a potential confounding variable, males and females were then re-analyzed independently. Similarly, two age-matched cohorts with a caliper distance of 0.3 were randomly created and re-analyzed. A standard multivariate regression to predict procedure group (ACLR vs. non-ACLR) was then completed with covariates of age, sex, and the eight SNAC questions. A 95% confidence interval in IBM SPSS 24 was used for all statistical tests (IBM Corp., Armonk, NY, USA).

## 3. Results

### 3.1. Procedure Group Comparison

Across 485 respondents, 243 (50.1%) patients underwent an ACLR surgery compared to 242 (49.9%) non-ACLR surgeries (Table 2). The ACLR group was older (*p* = 0.028) and more frequently post-menarche (*p* = 0.009). No difference was observed between groups in time between surgery and survey completion. Significant differences were found in the percentage of positive responses to half of the SNAC questions. Notably, ACLR respondents were more likely to report a change in appetite (*p* = 0.021) and a recent change in weight (*p* = 0.011). ACLR respondents also more often reported a desire for a better understanding of nutrition (*p* = 0.004) and a goal or recommendation to change their body composition or weight (*p* = 0.032). No significant differences between procedure groups were found in willingness to undergo a nutrition consultation, though a significantly greater percentage of ACLR patients were identified and, thus, prompted for a consultation (*p* = 0.002). Two females (0.9%) did not respond regarding period history, and ten females (4.7%) did not respond regarding period regularity (Table 2). The post-ACLR patients also tended to have more total positive responses per patient (Figure 1).

### 3.2. Influence of Sex and Age

After sex-controlled analysis, all nutritional habits observed more frequently in ACLR patients remained, though occasionally with less statistical significance. In females, ACLR patients remained more likely to desire an understanding of nutrition (42.7%) than non-ACLR patients (29.2%; *p* = 0.032) and more often reported an appetite change (20.9%) than non-ACLR patients (8.3%; *p* = 0.007). In males, ACLR patients remained more likely to undergo a change in weight (21.1%) than non-ACLR patients (10.7%; *p* = 0.024) and were more likely to be identified for a consultation (75.2%) than non-ACLR patients (60.7%; *p* = 0.013).

After age matching, 219 patients in each procedure group were analyzed (15.1 ± 2.0 years), and all statistically significant relationships among the broad cohort were again identified. ACLR patients more frequently reported recommended changes in weight (21.0%) than non-ACLR patients (13.7%; *p* = 0.044) and actual changes in weight (17.8%) than non-ACLR patients (9.6%; *p* = 0.012). ACLR patients also underwent appetite changes (15.5%) more frequently than non-ACLR patients (9.1%; *p* = 0.042) and had a greater desire to understand nutrition (49.8%) than non-ACLR patients (35.6%; *p* = 0.003). Finally, more ACLR patients were identified for a consultation (75.3%) than non-ACLR patients (62.6%; *p* = 0.004).

### 3.3. Multivariate Regression

A standard multivariate regression with an outcome variable of procedure group resulted in an R-squared value of 0.049 (*p =* 0.008). Sex and age were not determined to be significant covariates. Skipping meals and a desire for better understanding of nutrition were significant predictors, while changes in appetite approached statistical significance (Table 3).

## 4. Discussion

Balanced and effective nutrition is a key component of returning to sport in both psychological and physiological respects [11]. Among all patients who completed a nutrition questionnaire after surgery, the frequency of being identified for a nutrition consultation was high (68.2%). Those who underwent an ACLR were identified even more frequently. These results suggest that nutrition requires further emphasis in the peri-operative period, and such emphasis may be especially beneficial after procedures with long and intensive recovery times, such as ACLR, compared to other lower-extremity orthopedic procedures [12]. Despite high consultation identification, a low rate of consultation acceptance was observed (33.0%). To address the frequent desire of patients for a better understanding of nutrition, particularly among ACLR patients, coaches and other mentors with whom the player is already familiar may be an additional important avenue for nutrition education in those refusing nutrition consultations. Such education is essential given that long-term recovery and return-to-sport are dependent on nutrition to heal microtrauma experienced by the tissue even well into the rehabilitation process [13]. Certain micronutrients have been identified that correspond to anti-inflammatory and collagen-building improvements after muscle, bone, and tendon/ligament injuries [14], but consistent habits that decrease an athlete’s ability to recover may limit the efficacy of even the most nutrient-effective diets.

In the current study, post-ACLR patients identified more with habits related directly to caloric intake, such as weight changes, suggested weight changes, or appetite changes. While the SNAC survey did not specify whether actual or suggested weight changes referred to increased or decreased weight, the results postulate that post-ACLR patients suffer a greater disruption to their perception of a healthy weight than other lower-extremity procedures. Weight gain is a common occurrence after surgery due to paired immobilization and high energy requirements [15], which decrease muscle protein synthesis and increase anabolic resistance [16]. Immobilization-induced protein synthesis changes and anabolic resistance may be affected in merely days, though supplements have been shown to shorten recovery [17] and improve strength [18,19], particularly those with collagen, vitamin C, and glycine, which target ligament tensile strength [20] and the extracellular matrix [21]. Weight loss is also a concern given immobilization and muscle atrophy [9,10] coupled with decreased food intake. Regardless of the direction of these weight changes, post-ACLR patients may require more oversight as they abruptly stop exercise and then progressively increase activity through rehabilitation.

In contrast to the more frequent weight-related risks in post-ACLR patients, secondary outcomes of risky nutrition—fatigue, dizziness, and stress fractures, for example—were only nominally increased. This lack of difference may be due to the fact that patients were assessed at their first post-operative visit before any physical therapy or exercise. Without activity, they may not have had the opportunity to experience changes to these secondary outcomes. More importantly, these outcomes indicate poor nutritional habits, which existed prior to the injury, and can help dietitians anticipate how an athlete may be at risk after surgery. Future work may benefit from similar survey questions regarding fractures, fatigue, dizziness, and similar secondary outcomes at a point closer to clearance for return-to-activity as well as a comparison with objective dietary questionnaires both pre- and post-surgery. In the females of this cohort, post-ACLR patients were found to be more frequently post-menarche, though potentially a result of an older mean age in this group. Females in the ACLR group, though not statistically significant, also reported a higher rate of irregular periods. Nutrition has been linked to a missed or irregular period [22], and athletes are more likely to have irregular periods [23]. Thus, while delayed periods may not be a primary concern for post-ACLR patients, irregular periods may act as an indicator for healthcare professionals to monitor nutritional habits and consider inadequate fueling as both a potential cause and area for improvement.

Despite an overall reduction in statistical significance after sex-controlled analyses, trends of riskier nutritional habits in post-ACLR patients remained the same: Males and females both reported greater nutritional risks after ACLR. Among similarly aged patients with and without an ACLR, the current study suggests that those post-ACLR are more likely to report nutritional risks. Given the lack of an effect due to age-matching, it is likely that risky post-operative nutritional habits are not isolated to older athletes but are also present in younger athletes, which may be of greater concern during the periods of peak growth [24]. After regression analysis, a desire to understand nutrition significantly predicted being in the ACLR group, while skipping meals significantly predicted being in the non-ACLR group. Though not identified in the chi-square analysis, it is possible that patients who learn of an ACL tear diagnosis enter into a more rigorous preparation and physical therapy program than other diagnoses, which may be less severe, thereby skipping fewer meals, but this conjecture requires further study. Data on activity level or social concerns were not available in the current study but may have a measurable impact. Notably, periods of rest can introduce negative health outcomes both physically and psychologically among those who have been diagnosed with an injury and must abstain from activity [25,26]. In a previous analysis of the SNAC survey, it was found that 15% of post-surgical respondents were concerned with weight and 30% skipped meals regularly [27], which could be a partial effect of busy activity schedules or social perceptions of the ideal adolescent athlete body.

Accordingly, the limitations of the current study include a broad range of ages and activity levels as well as varying procedures in the non-ACLR group. However, the goal of this study was to examine differences between ACLR patients and the general lower-extremity surgical population due to the high prevalence of ACLR among youth sports surgeries. As the first study, to our knowledge, to systematically evaluate specific nutritional habits post-operatively, there is a benefit to providers to have a more generalized perception of the patient population. These limitations were further examined through age- and sex-controlled analyses, which reported similar findings to the broad study. The SNAC survey, while not validated, may also be able to offer providers insight into the psychological aspects of nutrition as a source factor, which can be modified through targeted education, rather than purely physiological measures, such as strength, quadricep atrophy, or body mass index. Finally, recovery time was likely longer in the ACLR group than general lower-extremity procedures, though recovery time was not monitored. However, the assessment of patients just over one week after surgery does mean that they are unlikely to have started rigorous physical therapy at that point, so post-operative activity levels are likely similar. Athletes would also be informed of their recovery time prior to surgery, which may have an impact on their nutritional habits as they prepare for longer immobilization or more intense rehabilitation and should be considered in future studies.

## 5. Conclusions

Overall, the results of this study identify a higher frequency of risky nutritional habits in post-ACLR patients in comparison to the general lower-extremity orthopedic surgical population. Patients frequently declined their recommended nutrition consultations, but no difference in those who accepted a consultation was observed between procedure groups. Specifically, ACLR patients more often reported issues with weight-related questions regardless of age, which is an important consideration for providers due to the increased physiological need for energy after injury and surgical intervention. Future studies should examine secondary outcomes of weight change (dizziness, fatigue, or stress fractures) at return-to-sport clearance and identify the direction of reported weight changes (increased or decreased weight). Sports surgeons should regularly discuss nutritional habits with their patients and look for unhealthy relationships between a patient and their weight, particularly while undergoing recovery after ACLR. In cases where concerns may arise, discussion with post-ACLR athletes regarding weight should be conducted thoughtfully, as many may already be sensitive to the effects of inactivity on their weight, body composition, and body image.

## Figures and Tables

**Figure 1 nutrients-16-02379-f001:**
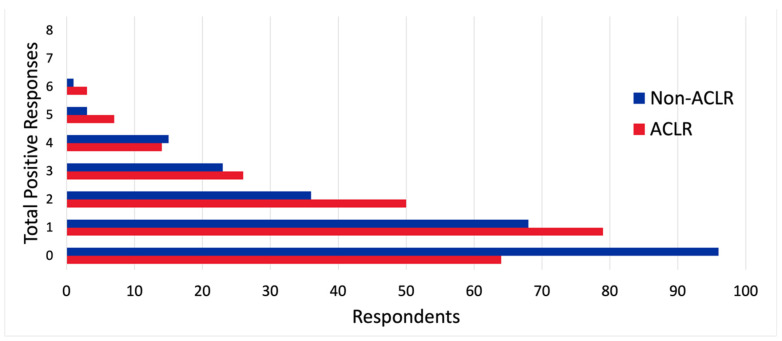
Total positive SNAC responses by procedure group. Note: Values represent the sum of positive responses to the eight SNAC questions in each group.

**Table 1 nutrients-16-02379-t001:** Sports Nutrition Assessment for Consultation (SNAC).

SNAC Question
Do you have any food allergies/intolerances or avoid any food groups?
2. Have you experienced any recent changes in appetite?
3. Do you regularly skip at least one meal a day?
4. Do you wish you better understood nutrition for your recovery?
5. Have you experienced any recent intentional or unintentional changes in your weight?
6. Are you trying to or has someone recommended that you change your body composition or weight?
7. Do you have a history of stress fractures?
8. Do you struggle with dizziness or fatigue during games, practices or with exercise?
If female:
9. Have you ever had a menstrual period?
10. Have you recently gone longer than 3 months without a period?
If any “Yes” response indicated to questions 1–8 or question 10:
11. In order to optimize your return to sport rehabilitation, would you like to meet with our sports dietitian?

Note: Questions assigned during intake of first post-operative visit.

**Table 2 nutrients-16-02379-t002:** Difference in SNAC responses between ACL and non-ACL groups.

Variable	ACL *N =* 243	NON-ACL *N =* 242	*p*-Value
Age (years)	15.3 ± 2.0	14.9 ± 2.1	**0.028 ***
Sex (% female)	110 (45.3)	120 (49.6)	0.341
Days to survey completion	9.1 ± 5.4	9.7 ± 5.0	0.076
Of the female respondents (% of females):			
Has had a period (*N* = 228)	107 (97.3)	104 (88.1)	**0.009 ***
Has recently not experienced a regular period (*N* = 201)	11 (11.1)	6 (5.9)	0.183
Sports Nutrition Assessment for Consultation			
Food allergies/intolerances	21 (8.6)	21 (8.7)	0.989
Changes in appetite	39 (16.0)	22 (9.1)	**0.021 ***
Skip meals	66 (27.2)	80 (33.1)	0.157
Desire for better understanding	117 (48.1)	85 (35.1)	**0.004 ***
Experienced changes in weight	41 (16.9)	22 (9.1)	**0.011 ***
Goal/Recommendation to change weight	51 (21.0)	33 (13.6)	**0.032 ***
History of stress fractures	12 (4.9)	9 (3.7)	0.509
Struggle with dizziness or fatigue	19 (7.8)	18 (7.4)	0.874
Consultation Outcome			
Identified for a consultation	181 (74.5)	149 (61.6)	**0.002 ***
Consultation accepted	64 (35.4)	45 (30.2)	0.321

Note: All values reported as *N* (%). Days to survey completion defined as days between surgery and completion of SNAC survey. Percentages are representative of the percent who responded within each procedure subgroup unless otherwise noted (e.g., % of females). Significant *p*-values in bold with an asterisk (*).

**Table 3 nutrients-16-02379-t003:** Multivariate regression for ACLR and non-ACLR groups.

Variable	Coefficient	Standard Error	*p*-Value
Constant	0.210	0.169	0.214
Sex	−0.022	0.047	0.634
Age	0.017	0.011	0.117
Food allergies/intolerances	−0.046	0.081	0.567
Changes in appetite	0.126	0.070	0.073
Skip meals	−0.129	0.051	**0.012 ***
Desire for better understanding	0.105	0.050	**0.035 ***
Experienced changes in weight	0.093	0.077	0.231
Goal/Recommendation to change weight	0.046	0.067	0.495
History of stress fractures	0.027	0.112	0.811
Struggle with dizziness or fatigue	−0.005	0.088	0.957
Results			
R-Squared	0.049		
Significance	**0.008 ***		
Observations	485		

Note: Coefficients reported as the unstandardized coefficient of standard multivariate regression model. Outcome variable is defined as the procedure group (ACLR or non-ACLR). Significant *p*-values in bold with an asterisk (*).

## Data Availability

Data are available upon reasonable request from the corresponding author. Data are not publicly available due to concerns for patient anonymity.

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
