# Peer review of "Greater Post-Operative Nutrition Risks Identified in Pediatric and Adolescent Patients after Anterior Cruciate Ligament Reconstruction Regardless of Age and Sex"

_nutrients, 2024, doi:10.3390/nu16152379_

Round 1
Reviewer 1 Report
Comments and Suggestions for Authors
In this manuscript, the authors presented the study on greater post-operative nutrition risks identified in pediatric and adolescent patients after anterior cruciate ligament reconstruction regardless of age and sex. This manuscript is interesting, there are some concerns before its publication for this journal.
1. Please provide the details on the construction and validation process of the questionnaire to ensure the reliability and validity of the results.
2. Is possible that the authors considered using multivariate regression analysis to control for potential confounding variables?
3. Can other potential confounders be considerate as potential confounders and performs subgroup analyses, like patients' levels of physical activity or dietary changes before and after surgery?
4. The discussion emphasizes the importance of nutrition in the recovery process. the authors should expand the discussion including nutrition's impact on long-term recovery and return to sports?
5. The author should provide the certificate approved by an ethics committee for this study?
Comments on the Quality of English LanguageNo
Author Response
General Comment: In this manuscript, the authors presented the study on greater post-operative nutrition risks identified in pediatric and adolescent patients after anterior cruciate ligament reconstruction regardless of age and sex. This manuscript is interesting, there are some concerns before its publication for this journal.
Response: Thank you very much for taking the time to review this manuscript. We have included detailed responses below, and the corresponding revisions/corrections are highlighted in the re-submitted files.
Point-by-Point Comments
Comments 1: Please provide the details on the construction and validation process of the questionnaire to ensure the reliability and validity of the results.
Response 1: Thank you for your comment. While we are currently in the process of validating the SNAC survey, the results are not yet published, therefore we cannot refer to the validation process. We acknowledge that this is a limitation of the study, and it has been mentioned in the limitations paragraph. We hope that the publication of this manuscript in addition to another currently in the literature will assist in forming a strong basis for the validation of the SNAC. We have also added another clarification to the inclusion of female-specific questions after the standard eight SNAC questions. If there are any further clarifications which can be made prior to its validation, we would be happy to make these changes.
Comments 2: Is possible that the authors considered using multivariate regression analysis to control for potential confounding variables?
Response 2: Thank you for this suggestion – we entirely agree that a multivariate regression would be beneficial. In addition to the statistics previously reported, we have run a multivariate regression to include in the results which should help assess potential confounders such as age and sex. Please find this addition at the bottom of the results section with the appropriate additions to the statistical section of the methods and the discussion.
Comments 3: Can other potential confounders be considerate as potential confounders and performs subgroup analyses, like patients' levels of physical activity or dietary changes before and after surgery?
Response 3: Thank you for your comment. Regrettably, we do not have an accurate assessment of physical activity or dietary changes available for the current patient cohort, though we now hope to collect these variables in the future. Given that patients were assessed just over one week after surgery and days between surgery and survey completion did not differ between groups, we believe post-operative physical activity differences were minimized. In this short period of time, patients were unlikely to start rigorous physical therapy, which also reduces the impact of varying rehabilitation times between the procedure groups. Still, we have introduced a comment on the lack of data for physical activity and dietary changes into the limitations given that there is a potential impact which was not assessed. Additionally, a note on future work has been added to the discussion. We also hope that the inclusion of a multivariate regression per your suggestion helps to account for potential confounders.
Comments 4: The discussion emphasizes the importance of nutrition in the recovery process. the authors should expand the discussion including nutrition's impact on long-term recovery and return to sports?
Response 4: Thank you for this suggestion. Additional discussion has been added to better discuss long-term recovery and return to sport in the context of nutrition. Please see the addition below:
“Such education is essential given that long-term recovery and return-to-sport are dependent on nutrition to heal microtrauma experienced by the tissue even well into the rehabilitation process [13]. Certain micronutrients have been identified which correspond to anti-inflammatory and collagen-building improvement after muscle, bone, and tendon/ligament injuries [14], but consistent habits which decrease an athlete’s ability to recover may limit the efficacy of even the most nutrient-effective diets.”
Comments 5: The author should provide the certificate approved by an ethics committee for this study?
Response 5: Thank you for your suggestion. The ethical approval number has been added to the manuscript, and an approval letter has been added to the submitted files.
Response to Comments on the Quality of English Language
Point 1: Minor editing of English language required
Response 1: We hope the clarifications included in this review satisfy the requirement for minor English language editing. Certain adjustments to word choice and typo corrections were made in this revised version, as well. All authors are native English speakers. If any edits are still required, we would be happy to make such adjustments!
Thank you again for taking the time to review this manuscript!
Reviewer 2 Report
Comments and Suggestions for Authors
The paper is potentially interesting and in line with this journal aims. Nevertheless, many concerns have to be addressed and clarified.
Introduction seems too little referred to the specific issue of this article, so the first part deals with general data about ACL reconstruction and it seems pleonastic, on the contrary too little is said about ACL recunstruction in young athletes.
Methods need a drastic improvement. Authors stated that: "After approval from the local Institutional Review Board according to the Declaration of Helsinki with a waiver of informed consent, patients who underwent lower-extremity surgery at a single tertiary sports medicine center were retrospectively reviewed. Patients were included if they were 8-19 years".
You should cite the ethical approval in this section (is it the following one STU 062017-100, approved 7/20/2017?). Why did you selected people aged between 8 and 19? In the title you spoke about pediatric and adolescent patients. Overall, it is not clear if you collected or not the informed consent. It is mandatory, especially in case of minor patients. And, as you know, the fact that it was a retrospective study did not exempt you from doing so.
In this sense, a clear definition of the study model is needed too.
The comparison between the groups is questionable. These groups are too heterogeneous, since the control one contains too many different diagnosis, resulting in many possible different surgical intervention and therefore different sequelae and rehabilitation periods, with different times which expose to different risks of heating disorders. Then, it is not clear if all patients were athletes or if they practice sporting activities at an amatuer level or if not. Did you deepen this?
Please clarify all these aspects to make the method of your study more solid.
As the discussion, it seems interesting but some further reasonings could be added. The eating habits of such a youg population could be affected also by other elements, such as sporting practice and possible social restrictions. You could extend discussion considering these aspects. TO do that, I suggest the following references:
- Farì G, Di Paolo S, Ungaro D, Luperto G, Farì E, Latino F. The impact of covid-19 on sport and daily activities in an italian cohort of football school children. Inter J [Internet]. 2021;26(5):274-8.
- Puia, A., & Leucuta, D. C. (2017). Children`s lifestyle behaviors in relation to anthropometric indices: a family practice study. Clujul medical (1957), 90(4), 385–391. https://doi.org/10.15386/cjmed-758
Best regard and good luck
Author Response
General Comment: The paper is potentially interesting and in line with this journal aims. Nevertheless, many concerns have to be addressed and clarified.
Response: Thank you very much for taking the time to review this manuscript. We have included detailed responses below, and the corresponding revisions/corrections are highlighted in the re-submitted files.
Point-by-Point Response to Comments
Comments 1: Introduction seems too little referred to the specific issue of this article, so the first part deals with general data about ACL reconstruction and it seems pleonastic, on the contrary too little is said about ACL recunstruction in young athletes.
Response 1: Thank you for your comment! We have adjusted the introduction to discuss less of ACLR in general and instead focus on youth ACL injuries, particularly how they are different from adults. Please find the new text below:
“Return-to-sport clearance for young athletes involves both functional clearance and psychological readiness [4,5] given that time off from sport may impact an athlete’s mental status [6]. Unlike adults, however, youth recovering from surgery must account for the energy needs of normal growth in addition to their physiologic requirements for recovery during a period in which body image and social consciousness are growing [7]. The culmination of these factors leaves youth recovering from ACLR or other surgeries in a vulnerable position regarding their rehabilitation success.”
Comments 2: Methods need a drastic improvement. Authors stated that: "After approval from the local Institutional Review Board according to the Declaration of Helsinki with a waiver of informed consent, patients who underwent lower-extremity surgery at a single tertiary sports medicine center were retrospectively reviewed. Patients were included if they were 8-19 years". You should cite the ethical approval in this section (is it the following one STU 062017-100, approved 7/20/2017?). Why did you selected people aged between 8 and 19? In the title you spoke about pediatric and adolescent patients. Overall, it is not clear if you collected or not the informed consent. It is mandatory, especially in case of minor patients. And, as you know, the fact that it was a retrospective study did not exempt you from doing so. In this sense, a clear definition of the study model is needed too.
Response 2: Thank you for your comment. Ethical approval has been cited in the methods to match the ethical approval statement at the bottom of the article. Additionally, it was noted why the study required a waiver of informed consent – we completely understand and agree that ethical approval is essential, particularly for the pediatric population. As this study aimed to investigate our patients’ survey outcomes retrospectively and data was de-identified immediately after collection, risks to the patient were low, and approval allowed for a waiver of consent. Therefore, while ethical approval was obtained prior to any study procedures, individual patients were not contacted for consent per the requirements of the Institutional Review Board.
Regarding the age range, patients aged 8-19 were chosen due to the limits of the institution’s patient population. Generally, surgeons at this institution do not regularly operate on those under eight years for lower-extremity sports surgery, and as a pediatric institution, patients over 19 are generally referred elsewhere. As such, these age limits include all but certain patients meeting an exception and allow us to most accurately represent our patient population. Furthermore, recent literature has explored ACL injuries at as young as eight-years old, suggesting that this age is appropriate to include in a pediatric lower-extremity surgery study. Please see the added text to the manuscript below:
“After approval from the local Institutional Review Board according to the Declaration of Helsinki (STU 062017-100), patients who underwent lower-extremity surgery at a single tertiary sports medicine center were retrospectively reviewed. As the study was retrospective and involved minimal risk to study subjects, a waiver of informed consent was obtained given the research could not practically be carried out otherwise. Patients were included if they were 8-19 years at the time of surgery given that younger patients are less frequently treated surgically at the institution. Included patients must have completed the institution-specific Sports Nutrition Assessment for Consultation (SNAC) between November 30th, 2021 – January 19th, 2023 at their first post-operative clinic visit.”
Comments 3: The comparison between the groups is questionable. These groups are too heterogeneous, since the control one contains too many different diagnosis, resulting in many possible different surgical intervention and therefore different sequelae and rehabilitation periods, with different times which expose to different risks of heating disorders. Then, it is not clear if all patients were athletes or if they practice sporting activities at an amatuer level or if not. Did you deepen this? Please clarify all these aspects to make the method of your study more solid.
Response 3: Thank you for this comment – we completely understand that using differing diagnoses could be questioned. Ultimately, the goal of our study was to examine how ACL diagnoses compare to other surgical interventions. To do this, we must use a heterogenous cohort of many different surgeries as a control group. To best offer this control, the cohort was reduced to only those with lower-extremity surgeries who presented to a single pediatric sports medicine center instead of a broader surgery type, age range, or type of patient (in terms of activity). Accordingly, while we do not have activity level data (though we now wish to pursue this in the future), we can be confident that the vast majority of these patients were athletes. Though we do wish we had data available on this breakdown of activity level, we believe that the heterogenous nature of the Non-ACLR group may actually present as a strength given our study’s purpose and the high prevalence of ACL injuries which need to be compared to other procedures.
Comments 4: As the discussion, it seems interesting but some further reasonings could be added. The eating habits of such a youg population could be affected also by other elements, such as sporting practice and possible social restrictions. You could extend discussion considering these aspects. TO do that, I suggest the following references:
- Farì G, Di Paolo S, Ungaro D, Luperto G, Farì E, Latino F. The impact of covid-19 on sport and daily activities in an italian cohort of football school children. Inter J [Internet]. 2021;26(5):274-8.
- Puia, A., & Leucuta, D. C. (2017). Children`s lifestyle behaviors in relation to anthropometric indices: a family practice study. Clujul medical (1957), 90(4), 385–391. https://doi.org/10.15386/cjmed-758
Response 4: Thank you for these comments and reference suggestions! Further discussion of this topic has been added with the above references. Please see the addition below:
“After regression analysis, a desire to understand nutrition significantly predicted being in the ACLR group, while skipping meals significantly predicted being in the Non-ACLR group. Though not identified in Chi-Square analysis, it is possible that patients who learn of an ACL tear diagnosis enter into a more rigorous preparation and physical therapy program than other diagnoses which may be less severe, thereby skipping fewer meals, but this conjecture requires further study. Data on activity level or social concerns were not available in the current study but may have a measurable impact. Notably, periods of rest can introduce negative health outcomes both physically and psychologically among those who have been diagnosed with an injury and must abstain from activity [25, 26]. In a previous analysis of the SNAC survey, it was found that 15% of post-surgical respondents were concerned with weight and 30% skipped meals regularly [27], which could be a partial effect of busy activity schedules or social perceptions of the ideal adolescent athlete body.”
In addition to the above comments, a number of clarifications and additions have been made to increase the quality of the manuscript, which you may find in the attached file. Thank you again for taking the time to review this manuscript!
Round 2
Reviewer 2 Report
Comments and Suggestions for Authors
Thank you for the efforts to improve the quality of your paper according to my suggestions.
The paper is now well structured, ehtical concerns have be clarified.
No further corrections are needed in my opinion.
Regards